# The Therapeutic Potential of Breast Milk-Derived Extracellular Vesicles

**DOI:** 10.3390/nu12030745

**Published:** 2020-03-11

**Authors:** Jeffrey D. Galley, Gail E. Besner

**Affiliations:** Center for Perinatal Research, The Research Institute at Nationwide Children’s Hospital, Department of Pediatric Surgery, Nationwide Children’s Hospital, Columbus, OH 43205, USA; jeffrey.galley@nationwidechildrens.org

**Keywords:** extracellular vesicle, exosome, necrotizing enterocolitis, breast milk

## Abstract

In the past few decades, interest in the therapeutic benefits of exosomes and extracellular vesicles (EVs) has grown exponentially. Exosomes/EVs are small particles which are produced and exocytosed by cells throughout the body. They are loaded with active regulatory and stimulatory molecules from the parent cell including miRNAs and enzymes, making them prime targets in therapeutics and diagnostics. Breast milk, known for years to have beneficial health effects, contains a population of EVs which may mediate its therapeutic effects. This review offers an update on the therapeutic potential of exosomes/EVs in disease, with a focus on EVs present in human breast milk and their remedial effect in the gastrointestinal disease necrotizing enterocolitis. Additionally, the relationship between EV miRNAs, health, and disease will be examined, along with the potential for EVs and their miRNAs to be engineered for targeted treatments.

## 1. Exosomes

Since secreted extracellular vesicles (EVs) were first discovered over 30 years ago [1,2], their involvement in the interrelated fields of physiology [3,4], immunology [5,6,7], and metabolism [8,9,10] has been the subject of extensive research. EVs are nanosized particles, between 30 and 2000 nanometers in size, that can be grouped based on their range of sizes. EVs between 30 and 150 nm are termed exosomes, while larger particles between 150 and 1000 nm are known as microvesicles [11]. The mechanism of formation for exosomes and microvesicles differ, in that exosomes are initially intraluminal vesicles, located in intracellular multivesicular bodies. As these bodies directly fuse with the parent cellular membrane, the smaller vesicles are exocytosed into the luminal space [12]. Microvesicles, the larger of the two, can fuse with the cellular membrane and directly bud from the cell [13]. Both exosomes and microvesicles are packaged with an assortment of molecules from the parent cell, including proteins, lipids, messenger RNA, and microRNAs [14,15]. The mechanism of formation of EVs as a whole is mediated by numerous proteins that eventually become part of the EV structure. These proteins, which embed in the EV bilipid layer, have been established as positive markers in EV confirmatory analysis, including tetraspanins such as CD63 and CD81 [16], vesicular fusion proteins such as flotillin [17], and the vesicular formation protein Alix [18].

EVs are released from multiple cell types throughout the body and participate in intercellular communication by fusing with recipient cell membranes and delivering the parent cell molecular cargo to the recipient target cell [19]. The functional outputs that occur as a result of EV-mediated shuttling of cargo are expansive. EVs can increase the proliferation and survivability of disparate cell types, including cartilage and endothelial cells, through modulation of AKT/ERK proliferative transcription pathways [20,21]. EVs can also influence innate and adaptive immunity, as demonstrated by their ability to modulate the NF-kB inflammatory signal transduction pathway, as well as T-cell priming and activation [22,23,24]. The source of the EVs has a major influence on their function. For example, stem cell (SC)-derived EVs have notable beneficial properties, including the ability to reduce the incidence of the severe intestinal disease necrotizing enterocolitis (NEC) in rat pups [25]. SC-EVs can also increase contractility and reduce diastolic pressure in murine hearts, accelerate mouse bone fracture healing, and increase neural growth—all hallmarks of beneficial health outcomes [3,26,27]. Additionally, EV cargo function is often analogous to the function of the parent cell from which the cargo originated. For example, EVs that are derived from immune cells such as dendritic cells are associated with regulation of the immune system [22], while EVs of cardiac origin mediate cardiac-related outcomes including anti-apoptosis of cardiomyocytes [28,29]. The cross-cell communicable functionality of EVs is also wide-ranging, with as many recipient cell types as there are cellular origins of EVs [30,31,32].

## 2. The Therapeutic Potential of Exosomes

The identification of EVs, and particularly of exosomes, as intercellular communicators that have far-reaching effects on human health has opened the door to a surfeit of potential uses for these nanoparticles. One common thread throughout exosomal and EV research is their therapeutic potential, and how they might be able to improve health outcomes systemically. The diverse molecules that compose EV cargo are the primary effectors of these therapeutic functions, and they can be surface-bound or found within the vesicle itself. Surface-bound bioactive molecules include antibodies, which can ameliorate complement-mediated cytotoxicity [33], and e-cadherin, a molecule involved in adhesion that can be bound to exosomes and utilized in tumor angiogenesis [34]. 

The internal cargo of EVs, including enzymes and miRNAs, impart therapeutic function on recipient cells upon delivery from the EV [35,36]. miRNAs, which are 18–22 nucleotide non-coding RNAs, have been identified as a primary player in EV therapeutic function by regulating gene expression post-transcriptionally [37]. The miRNAs form the RNA-induced silencing complex (RISC) in combination with the Argonaute silencing protein [38]. miRNAs have a specific seed sequence that targets a complementary sequence on the 3’-untranslated region of target mRNA [39]. At this point, the Argonaute protein represses translation by blocking formation of the translational protein complex or by actually degrading the mRNA [40]. Given the wide breadth of genes regulated by miRNAs in this manner [41], it is not surprising that exosomal/EV miRNAs have multiple natural therapeutic effects on health. For example, miRNAs delivered by secreted exosomes/EVs can enhance T-cell antigenic tolerance to contact sensitivity [42], increase neurite growth [43], and modulate the integrity of the brain vasculature by induction of endothelial cadherin [44], which maintains endothelial tight junctions. These miRNAs have also proven therapeutic in models of hepatitis, sepsis, and heart disease [45,46,47]. EV-derived miRNAs can have broad targeting effects on protein levels. Pro-inflammatory miRNAs, like miR-155, can increase pro-inflammatory IL6 protein levels in response to LPS when delivered via exosomes, while exosomal anti-inflammatory miR-146 reduce IL6 and IL12 protein levels while increasing the immunomodulatory cytokine, IL10 [48]. Innovations in exosome biology have allowed researchers to engineer exosomes/EVs for delivery of specific miRNA cargo [49,50]. Exosomes and miRNAs are highly stable and can withstand the stresses of the GI tract and the bloodstream, strengthening their case as a potentially tunable therapeutic system [51,52,53]. This can be done through transfection of a parent cell with an miRNA-mimic to over-produce a particular miRNA, or by transfecting exosomes/EVs directly, leading to an increase in miRNAs packaged within exosomes [35,49,50,54]. Electroporation, commonly used in the transformation or transfection of bacterial and mammalian genomes respectively, is often utilized to introduce over-expression of miRNA constructs into exosomes/EVs [49,50]. Upon in vitro synthesis, the exosomes can then be extracted, purified, and delivered therapeutically. In a study by Qu et al., adipose-derived mesenchymal stem cells were engineered to over-express miRNA-181-5p. Exosomes released from these stem cells were able to abrogate fibrosis in a liver disease model by increasing autophagy and inhibiting cell survival pathways [55]. A different study showed that over-expressed exosomal miRNA-181 reduced cardiac infarction and resultant inflammatory readouts including the IL-6 cytokine [56]. This can extend to other miRNAs, such as miR-134, which reduced breast cancer cell migration when over-expressed and released by EVs [57].

## 3. Disease and Exosomal miRNAs

Exosomes/EVs are not always associated with therapeutic outcomes. They have also been associated with disease states which has led to their cargo, especially miRNAs, being utilized as disease biomarkers for inflammatory diseases [58], neural disorders such as Alzheimer’s disease [59], and cancers such as leukemia and colon cancer [60,61]. Exosomal/EV miRNAs can also confer disease phenotypes, including transferring cancer drug resistance between cancer cells [62]. One such study demonstrated that sensitivity to tyrosine kinase inhibitors, a drug used against chronic myeloid leukemia, can be reduced by the export of miR-365 from resistant cells to sensitive cells [63]. Similar findings were reported by Wei et al., wherein exosomal release of miR-221/222 conveyed tamoxifen resistance to breast cancer cells [62]. Other studies have shown that exosomal miRNA-155 is inflammogenic through IL6 and IL8 upregulation [64], while exosomal miRNA-214-3p can inhibit bone development by interfering with osteogenic transcription factors [36], highlighting the wide range of deleterious health effects that exosomal/EV miRNAs are capable of mediating. Angiogenesis is associated with both negative health outcomes (e.g., tumor growth) and beneficial outcomes (e.g., wound healing and the resolution of cardiovascular disease), and EVs are capable of increasing angiogenesis [65,66,67]. Specifically, the transfer of miRNAs from exosomes/EVs has been associated with elevations in angiogenesis. miR-30b is one such miR that is transported via stem cell exosomes and can increase vessel branching via the inhibitory binding of DLL4, an anti-angiogenesis transcription factor [65]. 

In an extension of synthetically increasing miRNA expression as detailed above, researchers can also inhibit disease-associated miRNAs with anti-miR oligonucleotides, which are complementary to specific miRNAs and can bind and block their regulatory activity [36,68,69]. In one example, anti-miR-9 was constructed, and mesenchymal stem cells were seeded with the engineered anti-miRs. The authors showed that anti-miRs derived from these stem cells were able to inhibit miR-9-mediated resistance to the chemotherapeutic drug temozolomide, by increasing caspase activity [70]. Engineering elevations in miRNA production can also increase disease pathology. Over-expression of miRNA-212/132 increased the leakiness of the blood–brain barrier, while anti-miRs were able to abrogate these changes [71]. 

Pre-term birth is associated with multiple disease and adverse health outcomes, including necrotizing enterocolitis. Work is underway to examine how exosome/EVs and their miRNA cargo are involved in pre-term birth. One such study demonstrated considerable changes in the plasma-derived exosomal-miRNA profile of human mothers that delivered pre-term compared to those that delivered at term. Increased among the pre-term exosomal miRNA populations were miRNAs involved in glucocorticoid signaling, which is often elevated before labor progression [72]. Even non-EV-derived circulating miRNAs isolated as early as the first trimester can predict pre-term birth [73,74]. In mice, exosomal cargo composition shifts based on gestational age of the dam were observed, and proteomic analyses demonstrated that exosomes isolated from E18 dams were highest in inflammatory mediators. The authors of this paper surmised that this could be for cervical relaxation prior to birth. The importance of gestational age on exosomal composition was further established by the observation that E18 exosomes were capable of inducing pre-term birth, indicating that exosomes may be instrumental in the birthing process [75]. These studies offer compelling evidence for the wide-ranging effects that EV-miRNAs can have, acting as compositional biomarkers for pre-term labor and also being closely involved in the process of labor, pre-term or otherwise.

## 4. Breast Milk as a Therapeutic Bio-Fluid

Historically, the importance of breast milk (BM) in neonatal and infant health is unquestioned. The inherent benefits of BM are many and well-documented, including aiding in the development of the neonatal immune system through the delivery of antibodies like secretory immunoglobulin A [76], seeding and normalizing the infant microbiota as the BM is rich in beneficial bacterial species including Lactobacillus and Bifidobacterium [77], and reducing infant mortality in NEC and sudden infant death syndrome [78,79]. The reason for the multitudinous benefits of BM are due primarily to its composition. BM has a high quantity of proteins, many of which have immunological functions, including lactoferrin, immunoglobulin A, and cytokines [80]. Breast milk also contains milk oligosaccharides that are involved in infant immunity by strengthening gut barrier integrity [81], assisting in modeling health-associated microbiota [82], and activating production of the anti-inflammatory cytokine IL-10 [83]. Clinical reports also suggest that human milk oligosaccharide, whether supplemented in breast milk or formula, is protective against bronchitis and other lower respiratory tract infections [84].

Fatty acids (FAs) within breast milk are also impactful on neonate health, and their levels are differentially associated with disease prevalence. FAs are generally distributed between the saturated and unsaturated categories, and in total, make up nearly half of the energy supply for a breastfed infant. The major FA constituents of breast milk are of the polyunsaturated variety, including linoleic acid and its derivatives, arachidonic acid (ARA), and docosahexanoic acid (DHA). FAs have been associated closely with cognitive neurodevelopment and motor development. Breast milk is also a major source of vitamins for the newborn infant, with considerable concentrations of vitamins B, C, A, and D, as well as niacin and riboflavin [85]. While the mother’s diet is a major determinant of the levels of these vitamins, their prevalence in BM further establishes BM as an acutely important foundation of any infant’s diet.

## 5. Breast Milk Exosomes

In the search for exosomes/EVs that have a higher propensity for therapeutic effects, while also having the potential to be easily administered clinically, BM-derived exosomes have become a key player. Recently, there has been a spike in interest of the potential beneficial effects of BM exosomes [86,87]. These effects are spread across numerous mammals, including humans, bovines, camels, and pigeons [86,87,88,89]. A particular focus is being paid to the effect that BM exosomes have on the gastrointestinal (GI) tract, given recent studies that have shown that these human BM exosomes are resilient to digestion and can be endocytosed by intestinal epithelial cells, indicating that BM exosomes have the ideal qualities of a potential therapeutic. Resilience to digestion extends to BM EV-derived miRNAs [51,53,90]. In a study by Hock et al., rat BM-derived exosomes increased intestinal cell proliferation through the activation of stem cell marker, Lgr5 [87]. Similar findings were reported in a hydrogen peroxide model of oxidative stress. H2O2 increased cell death in the rat IEC-6 intestinal cell line. The addition of human BM-derived exosomes abrogated H2O2-induced increase in cell death, while significantly elevating cell viability over control [91]. 

The composition of BM-EVs is an open question. In a recent study, Wang et al. performed peptidomic analysis of BM exosomes, comparing mother’s milk from term pregnancies to mother’s milk from preterm pregnancies [92]. The authors discovered that preterm and term milk exosomes had altered peptide compositions comparatively. Cells treated with pre-term BM exosomes exhibited increased proliferation which was potentially associated with lactotransferrin and lactadherin, peptides that were up-regulated in the pre-term group [92]. Milk oligosaccharide has also been identified as a beneficial component in BM. In one study, the human milk oligosaccharide 2’-fucosyllactose protected against intestinal disease by increasing eNOS production. The elevated eNOS modulates mesenteric perfusion within the gut, a process that combats necrotic pathology [93].

Work is also underway to characterize the BM-EV miRNA profile, with miR-30d-5p, let-7b-5p, and let-7a-5p amongst the most abundant [94]. EV-miR-30d mediates the expression of genes involved in embryonic development, and let-7b is a well-known regulator of inflammation that mediates Toll-like receptor 4 activation, which can stimulate inflammation [95,96]. The composition of highly abundant EV-miRNAs is similar between term and pre-term milk, and both can be internalized by intestinal epithelial cells [51,90]. These miRNAs can then alter gene transcription within the cells. For example, internalized miR-148a reduces DNTM1 expression [97], while miR-22-3p, a highly expressed human BM-EV miRNA [51], downregulates NF-kB, a pro-inflammatory transcription factor [98]. As with protein composition, researchers have investigated how the “age” of the BM affects miRNA composition. One study reported that early milk in pandas, such as colostrum, contained miRNAs more closely associated with immune regulation and response, while later mature milk had more of a development and metabolism profile [99]. Interestingly, milk collected within the first week of a baby’s birth has a higher concentration of exosomes compared to milk collected by the baby’s second month of life [100]. 

As detailed above, breastfeeding has numerous beneficial effects, and in offspring it is negatively associated with obesity as well as reductions in exacerbations of asthma [101,102]. Given the correlation between BM-EV miRNA compositions and specific physiological functions including inflammation modulation and metabolism, it comes as no surprise that BM-EV miRNAs are likely to play a major role in preventing offspring disease prevalence. However, while these specific connections have not yet been made, it is generally known that exosomes derived from BM have broad immunomodulatory function, including reducing IL-2 and TNF-α production and inducing T-regulatory cell activation [103,104]. While deeper analysis tying BM-EV miRNAs to reduced risk of obesity or asthma development is necessary to draw conclusions, the breadth of beneficial outcomes mediated by BM-EVs gives credence to the hypothesis that BM-EV miRNAs are likely key mediators in most breastfeeding-associated outcomes.

The effect mediated by BM-derived exosomes can also be deleterious. Human BM exosomes are capable of inducing epithelial to mesenchymal transition (EMT), a phenomenon in which epithelial cells can lose their normal polarity and become increasingly migratory as the basement membrane breaks down [105]. This is often associated with the development of cancer, and cells that undergo this transition are generally invasive [106]. Additionally, mothers with Type-1 diabetes have an altered BM-EV miRNA profile, of which some of the differentially-expressed miRNAs are capable of inducing pro-inflammatory cytokine expression [107].

## 6. Breast Milk Extracellular Vesicles and Necrotizing Enterocolitis

BM-EVs have been targeted as a compelling therapeutic for NEC, a disease that affects premature infants, particularly in the low birth weight subset [108]. Early NEC symptomology includes abdominal distension, feeding intolerance, and ileus. In moderate cases, clinicians treat the disease with antibiotics, withholding of enteral feeds, and administration of total parenteral nutrition [109]. In severe cases, babies develop fulminant intestinal inflammation and eventual necrosis of the intestinal tissue. Mortality rates are as high as 40%–50% [108,110], and survivors can suffer from developmental disorders later in life, including memory deficits [111]. Infants that have undergone surgery may also have short bowel syndrome later in life, due to the reduction in overall intestinal length [109]. With these abundant issues associated with NEC, it is incumbent upon researchers to develop targeted, specific treatments for the disease. BM components, such as human milk saccharides (i.e., oligosaccharides like 2’-fucosyllactose and glycosaminoglycans like hyaluronan) and anti-microbial peptides (i.e., lactoferrin), have proven to have some efficacy in the treatment of NEC [93,112,113]. 

There is much interest in BM-EV therapeutic development for NEC. NEC animal models are widely used by research labs, including rat, mouse, and pig models [25,114,115,116,117]. Using the well-characterized rat NEC model, we have demonstrated that human BM-derived EVs reduce the incidence of NEC in rat pups when delivered orally with formula feeds. Orally delivered BM-EVs had improved efficacy compared to BM-EVs delivered intraperitoneally, which may be related to the stability of the BM-EVs trafficking through the digestive tract [51,86,90]. Human BM-EVs had pro-proliferative and anti-apoptotic effects in both rat (IEC-6) and human (FHS74) intestinal cell culture models, suggesting that the therapeutic properties of BM-EVs are mediated through a protective effect on intestinal epithelial cells [86]. These results corroborated other research demonstrating that BM-EVs have similar protective effects on cellular proliferation and growth [87]. We have also demonstrated that stem cell-derived EVs released from multiple sources, including amniotic fluid-derived mesenchymal stem cells (MSCs), bone marrow-derived MSCs, and amniotic fluid-derived neural SCs, also have therapeutic effects in NEC, significantly reducing overall NEC incidence in rat pups [25,115]. Both SC-derived EVs and SCs alone were associated with improved gut barrier function, abrogating the gut leakiness often associated with NEC, though the mechanistic involvement of SC-EVs in this process is presently unknown [115,118]. Protein production is likely one of the primary targets in EV/exosome-mediated therapeutics. BM-derived exosomes were able to mediate the expression of multiple proteins to reduce NEC severity, including reducing myeloperoxidase, a pro-inflammatory molecule released by neutrophils that is elevated in a mouse model of NEC, while increasing MUC2 and GRP94, which are expressed by goblet cells and are major components in the intact gut barrier [119]. In a murine LPS model, exosome administration reduced LPS-induced elevations in pro-inflammatory cytokine protein levels (e.g. IL-1, IL-6, TNF-α) [120]. In the same model, exosomes reduced pro-apoptotic caspase-3, as well as components of the pro-inflammatory signal transduction proteins, TLR4 and MyD88. Exosomes were also able to reduce LPS-induced phosphorylation of the NFκB pathway. Many of these anti-apoptotic and anti-inflammatory effects were mediated by exosomal miRNAs. Given previous studies that have demonstrated the tunable nature of EV miRNAs [35,36], BM-derived EVs continue to be a strong candidate in disease therapeutics. 

## 7. Future Directions of EV Therapy for NEC

Going forward, there is a crucial need for a better understanding of: (1) NEC pathophysiology, (2) NEC etiology, and (3) NEC inflammatory mechanisms. This knowledge will allow more stringent focusing of therapeutic EV cargos. As numerous studies have shown, NEC incidence and severity, as well as intestinal injury in commonly used in vitro models of gut disease, can be ameliorated through EV treatment [25,86,87]. The reductions in NEC injury severity by EVs may be mediated through EV-miRNAs, given their well-known ability to mediate disease outcomes [57]. Targeted treatments for these pathways, particularly through fine-tuning of EV-miRNA cargos, may prove to be an especially valuable avenue to pursue. Additionally, continued study on synthetically-produced EVs with specific pre-loaded cargos will provide members of the medical community an improved tool with proven ability to traffic to specific sites where their cargo can be unloaded.

## 8. Conclusions

This review has summarized recent research breakthroughs in the functionality of exosomes/EVs, particularly those derived from BM, as therapeutics in disease treatment. The fact that these EVs can store and transport stimulatory and regulatory elements, including miRNAs and enzymes, forms the basis for these nanoparticles to act as intercellular communicators as well as potential remedies for, or disseminators of, disease. Nearly every cell produces and exocytoses exosomes/EVs, including breast cells. BM is rife with exosomes/EVs, and given that the beneficial qualities of BM are well documented, it is of considerable interest to examine the extent to which these nanoparticles may mediate its positive effects. Herein, we have reviewed many of the recent studies that have investigated this very question, and we conclude that the evidence conveys strongly that BM-EVs are closely involved in health mediation. This is particularly observed in the treatment of necrotizing enterocolitis [86,119]. The use of BM-exosomes/EVs has proven to be a promising treatment in animal models of NEC, through pro-cellular proliferation and anti-apoptosis mechanisms [86]. However, further investigation is necessary prior to delivering these therapies clinically.

Researchers have discovered methods of amplifying or ablating EV miRNA composition, using silencing or over-expression vectors, and have demonstrated that such changes can have either deleterious or beneficial health effects [35,36,56,69]. These studies show promise for what can be performed with these particles and their cargos synthetically. The knowledge regarding the ability of these nanoparticles to be administered, to traffic to sites of disease, and to deliver their cargo, in combination with the capability to manipulate, tune, and improve their efficacy, creates a truly exciting opportunity in disease therapeutics.

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
