# Peer review of "The Therapeutic Potential of Breast Milk-Derived Extracellular Vesicles"

_nutrients, 2020, doi:10.3390/nu12030745_

Round 1
Reviewer 1 Report
The paper is well written and addresses an interesting topic. My major critique is that there was not enough attention given to the potential role of proteins in the review. I would say that if a section were added discussing more relevant protein targets the review could be accepted. A very good review but would make that correction to give a broader understanding of how these exosomes could exert their effect.
Author Response
Thank you for this critique. We have added sections on protein targets of exosomes and their cargo, with a focus on inflammation and gastrointestinal disease.
Reviewer 2 Report
Dear Authors,
I find the subject of your paper very interesting. The draft is well written.However the format of the text is more like a letter of editor than a systematic review.
If you want to write a review, I recommend following the PRISMA protocol.
The text lacks materials and methods, the search string, the comparison of the results of the different studies and the discussion of these.
Author Response
We conducted a general review on the therapeutic function of extracellular vesicles, with a focus on those derived from the breast milk. We did so by performing Pubmed searches of many search-strings including: breast milk exosomes, breast milk extracellular vesicles, extracellular vesicle miRNA, exosomal miRNA, therapeutic exosomes, therapeutic extracellular vesicles, extracellular vesicles disease, exosomes disease, exosome synthesis, extracellular vesicle synthesis, therapeutic breast milk, beneficial breast milk.
Reviewer 3 Report
The review article addresses the potential therapeutic benefits of exosomes and extracellular vesicles (EV) in breast milk. The topic is of high interest from a clinical and scientific point of view, as breast milk has many health-promoting effects, while the underlying mechanisms are not yet fully understood. The review article is very well written and a pleasure to read. Exosomes and EVs are introduced as intracellular communicators and carriers of active regulatory and stimulatory molecules (e.g. miRNAs and enzymes) and potential targets for preventive and therapeutic effects. Given the wide breadth of genes regulated by miRNAs delivered by secreted exosomes/EVs effects on sepsis prevention or avoidance of sustained inflammation in vulnerable newborn infants seem to be obvious.
I would recommend to add some data on the role of miRNAS in the context of term and preterm birth.
The review underlines the importance of state-of-the-art technology (e.g. engineering of exosomes) for scientific progress and nicely delineates a potential role of exosomal miRNAs in disease, as biomarkers or “disease phenotype (e.g. cancer). The authors have expertise in models of necrotizing enterocolitis (NEC), a devastating disease which can be prevented by human milk feeding. The NEC preventive role of orally fed exosomes from breast milk was demonstrated in rat pups which is mediated through a protective effect on intestinal epithelial cells. Apart from human milk, other sources of NEC-preventing exosomes might be stem cells in amniotic fluid and bone marrow-derived MSCs. Would the authors speculate on other aspects of breast-feeding which might be influenced by exosomes, e.g. risk of asthma, obesity?
In general, this review offers an excellent update and describes future directions of research on the functionality of exosomes and potential therapeutic strategies.
Author Response
- I would recommend to add some data on the role of miRNAS in the context of term and preterm birth.
Response: Thank you for this recommendation. We have added a section on the topic of how EV-derived miRNAs differ in the comparison of term and preterm birth.
- Would the authors speculate on other aspects of breast-feeding which might be influenced by exosomes, e.g. risk of asthma, obesity?
Response: The effect of breast milk-derived exosomes on offspring health are likely far-reaching, given the functional breadth of exosomal cargo, which can include miRNAs and enzymes. Thus, there is considerable potential for exosomes from the breast milk to have an abrogative effect on asthma or obesity development in later life. We have added a section to focus on this.
Round 2
Reviewer 2 Report
.